# Gastric Cancer: Epidemiology, Risk Factors, Classification, Genomic Characteristics and Treatment Strategies

**DOI:** 10.3390/ijms21114012

**Published:** 2020-06-04

**Authors:** Julita Machlowska, Jacek Baj, Monika Sitarz, Ryszard Maciejewski, Robert Sitarz

**Affiliations:** 1Center for Medical Genomics OMICRON, Jagiellonian University Medical College, 31-034 Kraków, Poland; julita.machlowska@gmail.com; 2Department of Human Anatomy, Medical University of Lublin, 20-090 Lublin, Poland; jacek.baj@me.com (J.B.); maciejewski.r@gmail.com (R.M.); 3Department of Conservative Dentistry with Endodontics, Medical University of Lublin, 20-090 Lublin, Poland; mksitarz@gmail.com; 4Department of Surgery, Center of Oncology of the Lublin Region St. Jana z Dukli, 20-090 Lublin, Poland

**Keywords:** gastric cancer, *H. pylori*, incidence, mortality, molecular markers, adjuvant chemotherapy, targeted therapy

## Abstract

Gastric cancer (GC) is one of the most common malignancies worldwide and it is the fourth leading cause of cancer-related death. GC is a multifactorial disease, where both environmental and genetic factors can have an impact on its occurrence and development. The incidence rate of GC rises progressively with age; the median age at diagnosis is 70 years. However, approximately 10% of gastric carcinomas are detected at the age of 45 or younger. Early-onset gastric cancer is a good model to study genetic alterations related to the carcinogenesis process, as young patients are less exposed to environmental carcinogens. Carcinogenesis is a multistage disease process specified by the progressive development of mutations and epigenetic alterations in the expression of various genes, which are responsible for the occurrence of the disease.

## 1. Introduction

The cancer development process is caused by both genetic and environmental factor influences. Around 50% of cancer incidents might be provoked by environmental agents, mostly dietary habits and social behavior. The development and progression of tumors is a multiannual and multistage process. Cancer usually occurs after 20–30 years of exposure to damaging carcinogenic agents. The possibilities of modern medicine allow for better recognition of most cancers, in their advanced stages, where among 50% of cases radical resection enables recovery.

Gastric cancer (GC) is a multifactorial disease, where many factors can influence its development, both environmental and genetic [1]. Current statistics display GC as the fourth leading cause of cancer deaths worldwide, where the rate of median survival is less than 12 months for the advanced stage [2]. Gastric carcinoma as a malignancy of a high aggressiveness with its heterogenous nature, and still constitutes a global health problem [3]. That is why alternative prevention, considered as a proper diet, early diagnosis and follow-up proper treatments, leads to the reduction of recorded incidents [4]. GC is rather rare and is not prevalent in the young population (under 45 years of age), where no more than 10% of patients are suffering from disease development [5,6,7,8,9].

The most popular classification of GC is the Lauren classification. According to this division, two subtypes of GC are displayed: intestinal and diffuse [10]. They present different characteristics, including clinical features, genetics, morphology, epidemiology and expansion properties. This division also has an impact on surgical decisions, regarding the range of stomach resections. The intestinal subtype encompasses tubular and glandular elements, with multiple degrees of differentiation. The diffuse subtype displays poorly cohesive single cells without gland formation [11,12]. Additionally, GC with signet ring cells is relatively prevalent, being classified as a “diffuse type” according to the Lauren classification [10]. Currently, signet ring cell carcinoma is described as a weakly cohesive type of cancer, consisting mostly of tumor cells with prominent cytoplasmic mucin and an eccentrically placed crescent-shaped nucleus [13]. Regarding the age at the diagnosis, GC is divided into early-onset gastric carcinoma (45 years or younger) and conventional GC (older than 45) [9,14].

Describing the pattern of signatures for GC development might be an important approach for better recognition of treatment strategies. To reveal these signatures, it is of great importance to find future appropriate therapies in personalized medicine.

## 2. Epidemiology and Risk Factors for Gastric Carcinoma Development

### 2.1. Incidence, Mortality and Geographical Variability

Every year, around 990,000 people are diagnosed with GC worldwide, of whom approximately 738,000 die [15]. GC is the fourth most common incident cancer and the second most common cause of cancer death [16].

GC incidence is different concerning sex and geographical variability. Men are two to three times more susceptible than women [15]. The incidence displays huge geographical diversity. It is noted that more than 50% of new incidents come up in developing countries. Areas with the highest probability for GC development encompass regions like Central and South America, Eastern Europe and East Asia (China and Japan). The low-risk regions include Australia and New Zealand, Southern Asia, North and East Africa and North America [17]. The five-year survival rate is mildly good only in Japan. In Europe, the ratio fluctuates between 10–30% [18]. The increased five-year survival rate is probably due to early diagnosis using the endoscopic examination method, which allows for the early detection and resection of cancer.

### 2.2. Trends

GC incidence rates have decreased in the last few decades in most parts of the world [19]. The decline in sporadic intestinal types of GC is observed, and the occurrence of the diffuse type GC has increased [20,21]. The rate of proximal GC is higher in comparison to the distal one. This trend might be explained by the increased standards of hygiene, better food conservation, a high intake of fresh fruits and vegetables and *Helicobacter pylori* eradication [22].

### 2.3. Risk Factors

Several factors have been noted to have a significant impact on the increased risk of developing GC, like family history, diet, alcohol consumption, smoking, *Helicobacter pylori* and *Epstein–Barr* virus (*EBV*) infections, which are summarized in Figure 1.

A family history of GC is also one of the most crucial risk factors [23]. However, GCs are mostly sporadic, around 10% display a familial aggregation [24]. Inherited GCs with a Mendelian inheritance pattern encompass less than 3% of all gastric carcinomas [25]. Hereditary diffuse gastric cancer (HDGC) is the most recognizable familial GC, which is caused by cadherin 1 gene (*CDH1)* alterations. The risk of gastric carcinoma in patients with a family history is around three-fold higher than among individuals without such a history [26]. The number of available studies on GC incidence and family history is rather low, the family history of individuals undergoing health check-ups has been noted for around 11% [27]. The ratio of GC with a family history is greater in Asian regions than in Europe and North America, however the frequency of HDGC, in comparison to the incidence of familial gastric carcinogenesis in Asia, is rather low [28]. Therefore, environmental agents, more than genetic alterations, can affect the development of familial GC in countries with an increased incidence of the disease.

The correlation between dietary factors and the risk of GC development has been broadly studied. The World Cancer Research Fund/American Institute for Cancer Research (WCRF/AICR) summarized that fruit and vegetables are protectors against GC development, whereas broiled and charbroiled animal meats, salt-preserved foods and smoked foods probably enhance GC progression [29]. Food carcinogens might interact with gastric epithelial cells and provoke changes in genes and their expression. Interestingly, a high intake of sodium chloride was described as devastating the gastric mucosa, promoting cell death and regenerative cell proliferation in animal models [30]. The dietary or endogenous role of *N*-nitroso compounds has been displayed to significantly increase gastrointestinal cancer risk, mostly among non-cardia GCs [31].

Among a variety of habits which play a role in GC development, the impact of smoking and alcohol intake has been considered. Studies show that smokers display around an 80% increase in the risk for GC development among non-drinkers. Additionally, heavy drinkers show a higher risk of GC; in a group of smokers, the risk of GC is estimated to be 80% [32]. In the European prospective nutrition cohort study, 444 cases of GCs were examined; heavy alcohol intake at the baseline was positively correlated with GC risk, whereas a decreased intake was not [33]. Intestinal non-cardia carcinoma was accompanied by heavy alcohol consumption. The dependence between alcohol intake and the risk of GC development was studied in a Korean population showing the *ALDH2* genotype [34]. Among a group of patients with *ALDH2**1/*2 carriers, current/ex-drinkers displayed a higher probability for cancer development in comparison to the group of never/rare drinkers. The study showed the association for alcohol consumption and GC development among a group of patients with *ALDH2* polymorphisms and the *ALDH2**1/*2 genotype.

*Helicobacter pylori (H. pylori)* is a Gram-negative bacterium that has been described as a class I carcinogen of GC development by the World Health Organization since 1994 [35]. The effect of *H. pylori* on the oncogenesis process has been described by two main mechanisms: an indirect inflammatory reaction to *H. pylori infection* on the gastric mucosa and a direct epigenetic outcome of *H. pylori* on gastric epithelial cells [36]. Several virulence factors of *H. pylori*, like CagA or VacA, are noted to increase the risk of GC development [37]. *H. pylori* with *cagA* and *vacA* relate to a higher risk of developing both intense tissue responses and premalignant and malignant lesions in the distal stomach [38]. Multiple epidemiological studies have shown that *H. pylori* infection is one of the risk factors of GC development. Besides, *H. pylori* infection impairs the gastric tissue microenvironment, promoting epithelial–mesenchymal transition (EMT) and further GC progression [39,40].

Apart from *H. pylori* infection, the second factor associated with GC development is the *Epstein–Barr virus* (*EBV*). *EBV* is a ubiquitous infectious factor. The *EBV* genome subsists in the tumor cells and transforming *EBV* proteins are expressed among them [41]. About 10% of GCs have been described to be *EBV*-positive, but there is not enough evidence for a distinct etiological role of *EBV* in GC development [42]. *EBV*-positive gastric carcinomas differ due to patients’ characteristics, like sex, age or anatomic subsite, and decrease with age among males [43].

## 3. Gastric Cancer Classification

### 3.1. Classification Systems in Gastric Cancer

In 1965 the Lauren classification of GC was established, and nowadays it is the most frequently used, compared to other available GC classifications [10]. According to the Lauren division, two histological subtypes of GC can be distinguished—intestinal and diffuse; later the indeterminate type was also included to characterize infrequent histology. Signet ring cell carcinoma is assigned to the diffuse subtype. Multiple studies have shown that the intestinal type is the most common, the second is diffuse and ending with the indeterminate type [10]. Intestinal carcinoma is characterized by visible glands and cohesion between tumor cells. The diffuse subtype encompasses poorly cohesive cells, diffusely infiltrating the gastric wall with little or no gland formation. The cells are usually small and round, also with a signet ring cell formation. There is evidence that the intestinal subtype is associated with intestinal metaplasia of the gastric mucosa and the occurrence of *H. pylori* infection. Some studies also revealed that the incidence of the diffuse GC subtype is higher among females and younger patients, and that this type of GC originates from the normal gastric mucosa [44].

The World Health Organization (WHO) classification issued in 2010 is perceived to be the most detailed among all classification systems. The WHO classification, apart from stomach adenocarcinomas, also describes other types of gastric tumors with decreased attendance [45]. The gastric adenocarcinoma type includes multiple subgroups, like tubular, mucinous, papillary and mixed carcinoma, which are similar to the indeterminate type according to the Lauren classification system. The poorly cohesive carcinoma type contains the signet ring cell carcinoma. The remainder of the classified gastric adenocarcinomas are described as uncommon, mainly because of their low clinical importance. Following the WHO classification, the most common GC subtype is tubular adenocarcinoma, then the papillary and mucinous types. The signet ring cell carcinoma encompasses around 10% of GCs and is described by the occurrence of signet ring cells in over 50% of the tumor [44,45,46,47].

GC development onsets are present in Figure 2, where the percentage of each carcinoma is displayed.

### 3.2. Conventional Gastric Cancer

Gastric carcinomas that appear intermittently mostly occur among the older population, at over 45 years of age, and are so-called “conventional gastric cancers”. The genetic factors that cause cancer development are less important in this type of cancer, where environmental agents are prevalent [48]. Patients are diagnosed between 60 and 80 years of age. These gastric carcinomas affect mostly men, who are two times more likely to develop them than women [49,50].

### 3.3. Early-Onset Gastric Cancer

Early-onset gastric cancer (EOGC) is described as a GC occurring at the age of 45 years or younger. Around 10% of GCs are categorized as EOGCs, however rates differ between 2.7% and 15%, depending on the performed cohort studies [14]. In the young population, diffuse lesions are more frequent and they are related to the background of the histologically “normal” gastric mucosa. Young patients are less exposed to environmental carcinogens, therefore, an EOGC is a good model to study genetic alterations in the gastric carcinogenesis process [51]. *H. pylori* infection is important for the development of tumors in EOGC patients, however, there is no statistically significant difference in the distribution of *IL1*β polymorphisms between young and old patients [9]. *EBV* infection is observed to be importantly decreased or absent in EOGCs [52]. It is postulated that around 10% of EOGCs have a positive family history [53]. It has been revealed that the early-onset type has different clinicopathological characteristics compared to the conventional subtype, which suggests that they display separate models within gastric carcinogenesis, and molecular patterns support this [54].

### 3.4. Gastric Stump Cancer

Gastric stump cancer (GSC) is described as a carcinoma in the gastric remnant after partial gastric resection, usually due to peptic ulcer disease (PUD). The incidence of GSC varies from 1% to 8% [55]. The major pathogenesis of GSC is biliary pancreatic reflux, provoking chronic inflammation of the mucosa, followed by atrophic gastritis, intestinal metaplasia and dysplasia. Other possible causes are achlorhydria and bacteria overgrowth, and *H. pylori* seem to be the main agent included in the etiopathogenesis of the GSC [56]. The surveillance of these patients with endoscopy and biopsies might allow for the early diagnosis of these patients, however, the benefit to cost ratio is still to be considered. Viste et al. (1986), made a comparison of GSC patients with other GC patients and discovered relevant differences in gender, age, staging, resectability rates and operative procedures, however, the postoperative mortality and survival rates were approximate [57].

### 3.5. Hereditary Diffuse Gastric Cancer

Hereditary diffuse gastric cancer (HDGC) is an autosomal dominant susceptibility for diffuse GC, a weakly differentiated adenocarcinoma that penetrates into the stomach wall leading to the thickening of the wall, usually without producing an explicit mass. The median age of HDGC onset is around 38 years, with a range of 14–69 years [58]. HDGC should be considered for screening with several important symptoms, like two or more documented cases of diffuse GC in first- or second-degree relatives, with at least one diagnosed before the age of 50, or three or more cases of documented diffuse gastric cancer in first/second-degree relatives, independent of the age of onset [59]. When clinical characteristics and family history are insufficient, the identification of a heterozygous germline *CDH1* pathogenic variant using screening with available genetic tests checks out the diagnosis and enables for family research [60,61].

Among *CDH1* mutation-negative patients within HDGC families, there were displayed candidate mutations within genes of high and moderate penetrance, like: *BRCA2*, *STK11*, *ATM*, *SDHB*, *PRSS1*, *MSR1*, *CTNNA1* and *PALB2* [62]. Therefore, in HDGC families, with no detected alterations in the *CDH1* gene, the clinical importance of other tumor suppressor genes, like *CTNNA1*, should be considered. *CTNNA1* is concerned in intercellular adhesion and is a questionable tumor suppressor gene for HDGC. The group discovered a novel variant (N1287fs) in the *BRCA2* gene, which is the first report of the occurrence of a truncating *BRCA2* variant among HDGC families. That is why it is important to consider HDGC syndrome as associated to *CDH1* mutations and closely related genes, then consider the clinical criteria of families with heterogeneous susceptibility profiles.

## 4. Genomic Characteristics of Gastric Cancer Development

Many studies on the molecular biomarkers of GC have been broadly investigated to reveal the wide spectrum of recognition patterns in this field. The main signatures for GC disease development encompass the modules of HER2 expression, factors that regulate apoptosis, cell cycle regulators, factors that influence cell membrane properties, multidrug resistance proteins and microsatellite instability [63], which are presented in Table 1.

### Possible Biomarkers of Gastric Cancer

Carbohydrate antigen 19-9 (CA 19-9) is the serum tumor marker most commonly used in cases of pancreatic cancer diagnosis or therapy monitoring. Physiologically, the serum concentration of CA 19-9 is small (less than 37 U/mL), being overexpressed in inflammatory conditions (e.g., pancreatitis) or other gastrointestinal diseases (esophageal, gastric or biliary cancers) [85]. The utility of CA 19-9 as a diagnostic biomarker of GC is slightly controversial and the results of the studies usually remain contradictory. Feng et al. reported that increased levels of CA 19-9 are associated with female gender and the presence of lymph node metastasis [86]. CA 19-9 might be associated with the tumor depth, tumor stage and lymph node metastasis in GC patients [87,88]. Besides, serum CA 19-9 levels are more diagnostically important than CEA regarding the estimation of the tumor size [89]. Serum levels of CA 19-9 are higher in GC patients compared to those with gastric benign diseases [90]. Increased CA 19-9 concentrations can also constitute a marker of an early recurrence after curative gastrectomy for GC, as well as of possible peritoneal dissemination [91,92]. Increased serum CA 19-9 and CA 72-4 levels are associated with an increased mortality rate among GC patients [93]. Song et al. reported that increased CA 19-9 levels are primarily observed in cases of stage III/IV group GC relative to the I/II group [94]. Usually, single tumor markers are not sufficiently sensitive and specific, therefore, the combined detection of several markers is inherent. In cases of GC, serum CA 19-9, carcinoembryonic antigen (CEA), carbohydrate antigen 72-4 (CA 72-4) and carbohydrate antigen 15-3 (CA 15-3) are important during an early GC diagnosis and therapy monitoring [95,96].

## 5. Prevention and Treatment Strategies

### 5.1. Prevention Strategies for Gastric Cancer

The two main primary prevention activities for gastric carcinoma at a population level could encompass a better diet habit and a lowering of the occurrence of *H. pylori* infection, the major cause of GC. The secondary prevention strategy is early detection using available resources, mainly the endoscopic method, as a gold standard.

### 5.2. Improvement in Diet

Prevention through dietary intervention might be possible through a higher intake of fresh fruit and vegetables and the restricted consumption of salt and salt-preserved food. Lifestyle modifications, including a higher level of physical activities and smoking limitation, could also reduce the risk of getting the disease. Fruit and vegetables are rich sources of folate, carotenoids, vitamin C and phytochemicals, which might have a protective role in the carcinogenesis process [97]. In the European Prospective Investigation into Cancer and Nutrition, 330 GC patients, both men and women, were examined [98]. A preventive role of vegetable consumption was displayed, mostly for the intestinal type of GC. Citrus fruit intake could play a role in protection against gastric cardia cancer. A subsequent report by the International Agency for Research on Cancer (IARC) described that the increased consumption of fruit “probably”, and higher intake of vegetables “possibly”, reduces the risk of GCs [99].

### 5.3. Helicobacter pylori Eradication

The prevention of GC development through *H. pylori* eradication is another approach. The explanation that the bacterium is a disease-causing factor allowed some authors, by 2005, to call for different programs to eradicate the infection among the population, as a way to limit the disease development [100]. A meta-analysis conducted by Ford et al. (2014) provides limited, moderate-quality proof that *H. pylori* eradication causes a reduction in the incidence of GC in healthy, asymptomatic, infected Asian individuals, however, these results cannot necessarily be extrapolated to various populations [101]. In the Shandong Intervention Trial, after two weeks of antibiotic dosing for *H. pylori,* the prevalence of precancerous gastric lesions decreased, while 7.3 years of oral supplementation with garlic extract, selenium and vitamins C and E did not [102]. In the prospective trial performed by Choi et al. 2014, the eradication of *H. pylori* after the endoscopic resection of GCs did not lower the incidence of metachronous gastric carcinoma [103]. Fukase et al. (2008) checked the prophylactic effect of *H. pylori* eradication on the development of metachronous gastric carcinoma after the endoscopic resection of early GC [104]. The study confirmed that the prophylactic eradication of *H. pylori* after the endoscopic resection of early GC should be used to prevent the development of metachronous gastric carcinoma. Although the randomized trials showed that *H. pylori* treatment might decrease GC incidence by 30–40%, there are still significant restrictions to the displayed data [104].

### 5.4. Early Detection Importance

The early detection of GC requires financial and population support, as well as available health services. Several tests are recommended and were used in various countries for GC screening. In Japan, mass screening for gastric carcinoma with a photofluorography method was started in 1960. Currently, over 6 million people are examined each year. The sensitivity and specificity of photofluorography are 70–90% and 80–90%, respectively. The five-year survival rate is 15–30% better among screen-detected cases than in symptom-diagnosed patients [105]. Additionally, endoscopic examination for gastric carcinoma has a higher sensitivity than the radiographic method [106]. The sensitivity of the endoscopic method in the population study was higher or the disclosure of distant or regional GC than for localized GC [106]. Upper gastrointestinal endoscopy has been established as the gold standard for the diagnosis of gastric carcinoma [107]. It is also performed for the minimally invasive treatment of early GC by endoscopic mucosal resection and endoscopic submucosal dissection. Matsumoto et al. (2013) performed the evaluation of the efficacy of radiographic and endoscopic examination for GC patients and suggested that both screening methods can allow for the avoidance of gastric carcinoma development [108]. Hamashima et al. (2013) investigated the evaluation of the reduction of mortality for GC patients by endoscopic examination. The results showed a 30% reduction in GC mortality using endoscopic screening in comparison to a control, the non-examined group, within 36 months before the date of diagnosis of GC [109].

### 5.5. Treatment Strategies for Gastric Cancer: Surgical Resection

Surgery plays a crucial role as a strategy in the treatment of GC [110]. The best time for surgery is when a tumor is mostly sensitive to the chemotherapy. The development of two new methods, endoscopic resection and minimally invasive access, have had an important impact on the treatment strategies revolution in the last few decades [111] Nevertheless, vertical and horizontal margin invasion and the chance of nodal implication should also to be taken under serious consideration to prevent real oncological lapses. The standard treatments are directed to the endoscopic mucosal resection, or, even better, endoscopic submucosal dissection (ESD) for differentiated types of gastric adenocarcinoma without ulcerative findings [112]. Both endoscopic mucosal resection and ESD provide favorable long-term outcomes. Laparoscopic surgery of GCs, as a minimally invasive method, was originally limited to treat distal-sided early GCs, with no necessity for complete gastrectomy or extended lymphadenectomy [113]. Both laparoscopic and robotic-assisted gastrectomies are considered to provide positive clinical outcomes, equivalent to those in cases of open surgeries. Furthermore, compared to open surgeries, minimally invasive techniques have even lower rates of postoperative complications, such as incisional hernias or bowel obstructions [114,115,116]. Limited surgical approaches—pylorus-preserving gastrectomy, proximal gastrectomy and local resection—significantly reduce the resection area of the stomach, as well as the extent of nodal dissection [117]. Conversion therapy in GC is an application of either chemotherapy or radiotherapy followed by surgical treatment in cases of originally unresectable or marginally resectable GCs, the application of which might be of great importance, especially in cases of stage IV GCs. [118]. Comprehensive surgical resection with lymphadenectomy D2 still constitutes the major treatment strategy aimed at cure for GC. The continuation of chemotherapy is usually crucial after the resection, preventing adverse events. Several reconstruction methods, such as Billroth I gastroduodenostomy, Billroth II gastrojejunostomy, casual/uncut Roux-en-Y gastrojejunostomy and jejunal interposition are often employed after the subtotal gastrectomy [119].

### 5.6. Adjuvant Chemotherapy

In the last few decades, multiple phase III trials have been undertaken to consider the potential of adjuvant chemotherapy versus surgery, however, no consistent outcomes have been observed [120,121,122,123]. The observations might be explicated by several important factors, like the huge heterogeneity of the study cohort, a low number of performed series, various levels of surgical precision and dissimilar chemotherapy regimens. A meta-analysis study, performed by the GASTRIC group in 2010, showed that postoperative adjuvant chemotherapy based on fluorouracil regimens significantly reduces the mortality rate of GC patients in comparison to surgery alone [124]. Adjuvant chemotherapy was correlated with a statistically important benefit in terms of overall survival and disease-free survival. There was no distinct heterogeneity for overall survival across randomized clinical trials. Five-year overall survival increased from 49.6% to 55.3% with chemotherapy. An application of oral fluoropyrimidine might also be effective in cases of advanced GCs [125,126]. Likewise, other phase III trials, including the CLASSIC or the ACTS-GC, proved that postoperative adjuvant therapy following D2 gastrectomy is a highly effective treatment strategy [127,128]. An activity of pembrolizumab in the neoadjuvant setting provides a rationale for its application in combination with chemotherapy in patients with resectable GCs [129]. The systematic review and meta-analysis performed by Yan et al. (2007) was undertaken to check the efficiency and safety of adjuvant intraperitoneal chemotherapy for patients with locally advanced resectable GC [130]. The study displayed that hyperthermic intraoperative intraperitoneal chemotherapy (HIIC), with or without early postoperative intraperitoneal chemotherapy (EPIC) after the resection of advanced gastric primary cancer, is assigned to increase the overall survival rate. Unfortunately, higher risks of intra-abdominal abscess and neutropenia are also displayed. Adjuvant XELOX might be a valid approach in curable gastric carcinomas among Asian patients. Nowadays, it is clear that adjuvant chemotherapy brings a survival benefit in radically resected GC for stage ≥ T2 or N+ [131,132]. Neoadjuvant chemotherapy followed by surgery is also highly recommended in cases of limited metastatic GCs [133]. What is also crucial while applying neoadjuvant chemotherapy is the genotype of the GC, which might additionally constitute a prognostic or predictive factor of the clinical outcome.

### 5.7. Neo-Adjuvant Chemotherapy

The importance of neoadjuvant chemotherapy in GC, gastroesophageal junction and lower esophageal adenocarcinoma has been highlighted over the past few decades. In the first Dutch randomized controlled trial of neoadjuvant chemotherapy, patients with proven adenocarcinoma of the stomach were randomized to obtain four series of chemotherapy with 5-fluorouracil, doxorubicin and methotrexate (FAMTX) prior to surgery or to undergo surgery alone. With a median follow-up of 83 months, the median survival after randomization was 18 months in the FAMTX group, versus 30 months in the surgery alone group [134]. In European regions, perioperative chemotherapy has been advertised based on the MAGIC [135] and FFCD9703 [136] randomized trials. In the first trial, Cunningham et al. (2006) investigated tests with epirubicin, cisplatin and infused fluorouracil (ECF) on patients’ survival with incurable locally advanced or metastatic gastric adenocarcinomas. Among a group of patients with operable gastric or lower esophageal adenocarcinomas, a perioperative regimen of ECF caused a lowering in tumor size, stage and importantly benefited progression-free and overall survival [135]. Boige et al. (2007) used the combination of 5-Fluorouracil (5FU) in a continuous infusion and cisplatin (FP) as one of the important approaches for advanced adenocarcinoma of the stomach and lower esophagus (ASLE). Preoperative chemotherapy using 5-fluorouracil/cisplatin improved the disease-free and overall survival of patients with ASLE [136]. Radiation therapy uses high-energy rays or particles to kill cancer cells. It is sometimes applied to treat stomach cancer. In the majority of cases, radiation therapy is given with chemotherapy (chemoradiation). Both neo-adjuvant chemoradiation therapy and neo-adjuvant chemotherapy significantly improve the clinical outcomes of patients with resectable GC with a similar efficiency [137].

### 5.8. Targeted Therapy

The major therapeutic options, based on the molecular characteristics of the gastric tumor, are ramucirumab and trastuzumab (targeting *VEGFR2* and *HER2*, respectively) [138]. Gastric cancer often displays heterogeneity of the *HER2* genotype and phenotype, which might be partly accountable for testing inaccuracy. Phase II trials studied trastuzumab plus chemotherapy (cisplatin, capecitabine) versus chemotherapy alone in *HER2+* advanced gastric patients and underlined that trastuzumab is the most appropriate therapeutic approach for strongly *HER2+* patients [139,140]. Other studies suggested that lapatinib, as a single targeted therapy, is weakly effective against gastric cancer, which might be explained by the contribution of antibody-dependent cell-mediated cytotoxicity (ADCC), which is lacking in the small molecule therapeutic approach [141]. Pertuzumab is another *HER2* monoclonal antibody that interacts with *HER*2 heterodimerization with different members of the *EGFR* family [142].

The epidermal growth factor receptor (EGFR) is amplified in approximately 5% of gastric cancers, specified by poor prognosis. Experiments have displayed a positive correlation between EGFR overexpression and cetuximab response [143]. A phase II trial assessing cetuximab plus oxaliplatin/leucovorin/5-fluorouracil displayed a dependence between a higher *EGFR* copy number and overall survival [144].

*VEGF*/*VEGFR2*-dependent signaling is significant in tumor angiogenesis. It has been noted that among GC cases, *VEGF* status and serum levels correlated with advanced stage and poor prognosis [145]. The role of ramucirumab, a *VEGFR-2* mAb, was evaluated in the REGARD study, as a second line therapy after disease progression on a first line chemotherapy regimen, among cases with unresectable, advanced gastroesophageal tumors [146]. A phase III study (RAINBOW) tested this antibody, in combination with paclitaxel, as a second line treatment among cases with metastatic GC who progressed after a first line chemotherapy [147]. Overall survival was importantly increased in the paclitaxel plus ramucirumab group in comparison to the placebo.

The fibroblast growth factor 2 receptor tyrosine kinase (*FGFR-2*) is overexpressed among approximately 10% of gastric tumors and its amplification is related to lymphatic invasion and poor prognosis [148]. Clinical trials in which patients picked for *FGFR2* amplification are treated with inhibitors, such as dovitinib or AZD4547, are ongoing [149]. The activation of the *PI3K*/*AKT*/*mTOR* pathway is often among GC tumors. A phase III clinical study investigated the *mTOR* inhibitor (everolimus) in patients with advanced gastric cancer, and the results showed no improvement in the overall survival [150]. Additionally, a phase II study of MK-2206, an inhibitor of *AKT*, displayed no positive results [151].

### 5.9. Imaging Strategies

Gastric cancer requires multimodal staging approaches, in which computed tomography (CT) is the first staging modality, mostly because of its broad availability and proper accuracy [152]. This method is very often used to assess local tumor invasion. It allows for poor soft tissue contrast; the intravenous contrast material and exposure to radiation is needed. Computed tomography for overall T-staging displayed a diagnostic accuracy between around 77% and 89% [153]. CT is frequently applied to image the occurrence of lymph node metastases among GC patients. The sensitivity was assessed as being between 63–92% and the specificity between 50–88%, according to a systematic review covering 10 studies [154]. The method of choice for M-staging is a CT of the abdomen and pelvis [155]. The sensitivity for the imaging of M1 disease using CT is approximately between 14–59%, and the specificity is between 93–100% [156].

Magnetic resonance imaging (MRI) is an auspicious method for depicting various gastric wall layers and the differentiation of tumor tissue from fibrosis [157]. The accuracy for the proper evaluation of the T-stage is between 64–88% [158]. MRI in T-staging was compared with CT, and the accuracy was rather higher for MRI, however, this difference was only proven to be statistically significant in two studies: 73% for MRI versus 67% for helical CT [159] and 81% for MRI versus 73% for spiral CT [160]. The precision of MRI for the correct distinction between node-negative and node-positive cases with GC varied between 65% and 100%, sensitivities and specificities ranged between 72–100%, 20–100%, 69–100% and 40–100%, respectively [161]. MRI is broadly applied to the diagnosis of liver metastases, as well as displaying capability for the diagnosis of peritoneal seeding [162]. The treatment response evaluation and the detection of lymph node metastases could take advantage of imaging biomarkers derived from functional MRI in the future [163].

Positron emission tomography (PET) imaging is not the best option for the evaluation of the T-stage. The resolution of PET is limited by the volume averaging of the metabolic signal, with prominent uptake averaged across several millimeters [164]. PET might be a very good method to detect anatomically small and metabolically active focuses of metastatic disease. The comparatively poor spatial resolution of PET causes the decreased productivity of differentiation compartment I and II nodes from the primary tumor itself [165]. PET is probably the most useful for the detection of distant areas of solid organ metastases. Kinkel et al. (2002) performed a metanalysis and underlined PET as the most sensitive noninvasive imaging strategy in this field [166]. PET may be a useful tool to prefigure answers to preoperative chemotherapy in GC cases.

## 6. Conclusions

In this review, we described GC characteristics, considering the epidemiology, risk factors, classification and molecular and genomic markers, as well as treatment strategies. We characterize the incidence of GC, which is variable when taking into account the geographical and sex variability. We displayed that the decline in sporadic intestinal types of GC is present, whereas the diffuse type prevalence is increased, and the proximal GC prevalence is higher than for the distal one. Several risk factors with an important impact on developing GC are mentioned, including family history, diet, alcohol consumption and smoking, as well as *Helicobacter pylori* and Epstein–Barr virus infection. The two main classifications of GC are described: Lauren, which is the most commonly used, and WHO, which is perceived to be the most detailed among all of the pathohistological classification systems. The signatures, which are described, are based on the current literature and research performed on this topic, which encompass: the module of HER2 expression, factors that regulate apoptosis, cell cycle regulators, factors that influence cell membrane properties, multidrug resistance proteins and microsatellite instability. We highlighted the two main primary prevention strategies for gastric carcinoma, which are better diet habits and a lowering of the occurrence of *H. pylori* infection, and the secondary prevention approach, which is early detection using the endoscopic method as a gold standard. Different treatment strategies are also displayed, including surgical resection, adjuvant and neo-adjuvant chemotherapy, radiation therapy, hyperthermic intraperitoneal chemotherapy (HIPEC) and pressurized intraperitoneal aerosol chemotherapy (PIPAC).

## Figures and Tables

**Figure 1 ijms-21-04012-f001:**
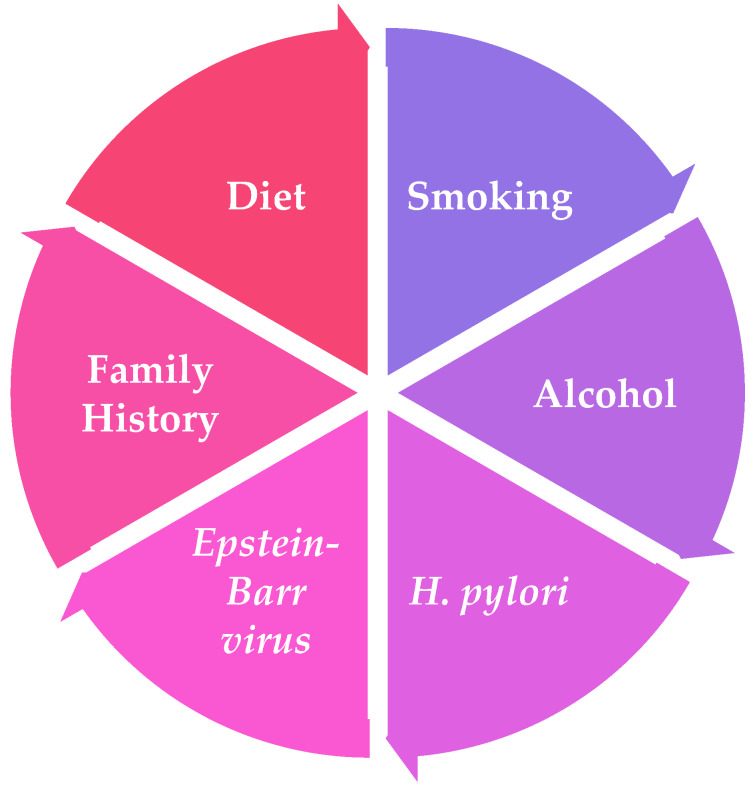
Risk factors for GC development.

**Figure 2 ijms-21-04012-f002:**
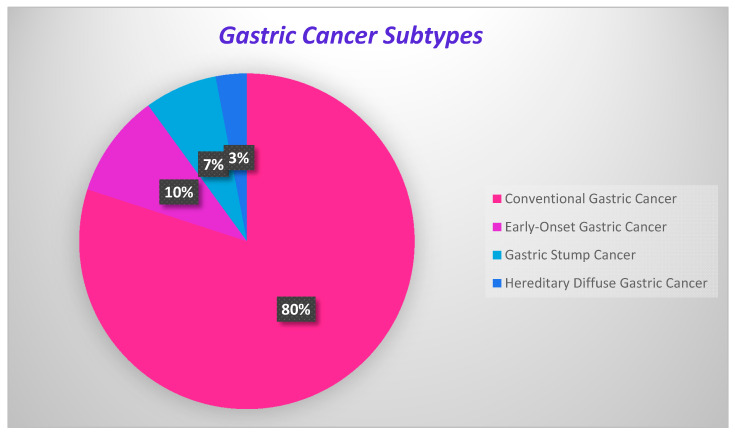
Onsets of gastric cancer development.

**Table 1 ijms-21-04012-t001:** Molecular biomarkers in gastric cancer development.

Molecular Biomarker	Impact on Gastric Cancer Development	Authors
***HER2***	-Amplification and overexpression in GC, the positive cases range from 6% to 30%. *-HER2/neu* amplification is higher in the intestinal histologic subtype of GC, compared to the diffuse subtype, and is not associated with gender and age, but with the poor survival of GC patients.	[64,65]
***p53***	-Mutations in the *p53* gene occur in the early stages of gastric carcinoma, and their frequency is increased in advanced stages of cancer development. -*TP53*-positive patients are also classified as one of the GC subtypes.	[66,67]
***PD1***	-The expression of *PDL1* is significantly increased in cases with PCNA and C-met expression, *EBV*-positive, and without metastasis; a better outcome is associated with increased *PD-L1*/*PD-1* expression.	[68]
***p73***	-The *p73* gene is not an object of genetic modification in gastric carcinogenesis, wild-type *p73* is quite often highly expressed in GC tissues by transcriptional induction of an active allele or the activation of a silent allele.	[69]
***mdm2***	The expression level of the MDM2 protein is importantly increased in intestinal metaplasia and gastric carcinomas in comparison to simple intestinal metaplasia and chronic gastritis.	[70]
***Bcl-2***	Lymph node metastases, depth of invasion and the negative expression of *Bcl-2* are associated with an increased chance of cancer recurrence.	[71]
***pRb*** ***CCND1***	-Cyclin D1 is a positive regulator of the cell cycle process; retinoblastoma protein (pRb) acts as cell cycle repressor, it promotes G1/S arrest and growth restriction through the inhibition of the E2F transcription factors; their higher expression is merged with cell overgrowth and cancer development. -The expression of pRb and cyclin D1 might be present in the early stages of gastric carcinogenesis, with the higher expression of Rb and cyclin D1 among nonneoplastic mucosa comprising dysplasia, intestinal metaplasia, atrophy and gastritis to carcinoma.	[72,73]
***p16***	The *p16* gene plays a main role as a tumor suppressor gene, the deletion of the *p16* gene is associated with the carcinogenesis process, as well as the progression of gastric carcinoma.	[74]
***p27^Kip1^***	Cyclin-dependent kinase inhibitor 1B, called p27^Kip1^ with low protein expression in GC, is assigned to advanced tumors, it is importantly higher in weakly differentiated cases and is described as a negative prognostic factor for the survival of patients.	[75]
***MUC***	Mucins are a group of extracellular, huge molecular weight, strongly glycosylated proteins; they have significant characteristics assigned to cell signalling, the creation of chemical barriers, facilities to create a gel, a major function related to lubrication. One of their main roles is also as an inhibitory function, and the high expression of mucin proteins, like MUC1, MUC2, MUC5AC and MUC6 is associated with gastric carcinogenesis process.	[76,77]
***MRP2***	The overexpression of MRP2 is significant in the initial absence of reaction to chemotherapy treatments of tumors, which allow us to consider it as an important biomarker for chemotherapy response.	[78]
***MDR1***	*MDR1* is a very significant candidate gene in the progress of GC susceptibility, as well as displaying an important impact on drug resistance response, and the knockdown of MDR1 might reverse this phenotype among GC cells.	[79,80]
***GST-P***	The expression of GST-P is visibly increased in tumors that are chemically induced, it is also associated with tumor invasion and recurrence, as well as poor prognosis.	[81,82]
***MSI***	-Microsatellite instability (MSI) is an important indicator of the DNA mismatch repair deficiency, which is an agent in the higher accumulation of genetic alterations in gastric carcinogenesis; MSI-positive patients do not have a high content of targeted mutations, some of them were detected in *PIK3CA*, *EGFR*, *ERBB3* and *ERBB2* genes. -GC cases with a high MSI can have long-term survival, regardless of the positive resection margin status.	[83,84]

Abbreviations: *HER2*—tyrosine kinase-type cell surface receptor, *p53*—tumor protein p53, *PD-1*—cell surface receptor programmed death-1 and its ligand (*PDL1*), *p73*—tumor protein p73, *mdm2*—murine double minute gene 2, *Bcl-2*—B-cell lymphoma 2, *pRb*—retinoblastoma protein, *CCND1—*cyclin D1 gene, *p16*—cyclin dependent kinase inhibitor 2A, *p27^Kip1^*—cyclin-dependent kinase inhibitor 1B, *MUC*—mucin, *MRP2*—multidrug resistance-associated protein 2, *MDR1*—multidrug resistance 1 gene, *GST-P*—glutathione S-transferases Pi, *MSI*—microsatellite instability, *PIK3CA—*phosphatidylinositol-4,5-bisphosphate 3-kinase catalytic subunit alpha, *EGFR –* epidermal growth factor receptor, *ERBB3—*Erb-B2 receptor tyrosine kinase 3*, ERBB2—*Erb-B2 receptor tyrosine kinase 2.

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
