# Peer review of "Gastric Cancer: Epidemiology, Risk Factors, Classification, Genomic Characteristics and Treatment Strategies"

_ijms, 2020, doi:10.3390/ijms21114012_

Round 1
Reviewer 1 Report
The authors summarized the current knowledge of gastric cancer and present a multi-faceted view of this disease. The review is well written and fits the scope of the International Journal of Molecular Sciences. Nonetheless, the relevance and contribution to the field are reduced by the fact that this review continues a spree of gastric cancer review by the same authors with naturally overlapping pieces of information (e.g. Gastric Cancer: Epidemiology, Prevention, Classification, and Treatment; Sitarz et al., 2018 PMID: 29445300 etc.). Still, the current article contains some updated information on the topic.
Specific points to address and suggestion to the authors:
1.) It would be beneficial to expand the section dedicated to familiar GC. While CDH1 represents a typical cancer predisposition gene mutated in families with HDGC, there are also familiar cases with unaffected CDH1. Several others HDGC predisposition mutations within genes of high and moderate penetrance were suggested - e.g. CTNNA1, BRCA2, STK11, SDHB, PRSS1, ATM, MSR1, PALB2 (PMID: 26182300).
2.) ALDH2 polymorphisms were reported to affect the development of gastric cancer associated with alcohol intake (e.g. PMID: 21507992), this could be discussed in the appropriate section.
3.) In Table 1. authors claim that "cyclin D1 and retinoblastoma protein (pRb) are positive regulators of the cell cycle process". This is not true for pRb. pRb acts as cell cycle repressor, it promotes G1/S arrest and growth restriction through inhibition of the E2F transcription factors. Please, correct this statement.
4.) The choice of graphical representation of diagram 2. is a bit unclear to me. What should the arrow represent? The age of onset? But then it would make more sense to have the order in the other direction (youngest patients on the left, oldest on the right - and then please label somewhere that the arrow represents the age dimension). From the first glance on the arrow and without prior knowledge, the graphics more intuitively, albeit wrongly, suggest that conventional GC can somehow "progress" into early-onset GC. etc. Possibly authors could consider to include a more intuitive diagram.
Author Response
Dear reviewer,
Thank you for a detailed review of our manuscript , " Gastric Cancer: Epidemiology, Risk Factors, Classification, Genomic Characteristics and Treatment Strategies". The manuscript has been improved taking into account the comments in your review. Our response point by point is given below.
Specific points to address and suggestion to the authors:
1.) It would be beneficial to expand the section dedicated to familiar GC. While CDH1 represents a typical cancer predisposition gene mutated in families with HDGC, there are also familiar cases with unaffected CDH1. Several others HDGC predisposition mutations within genes of high and moderate penetrance were suggested - e.g. CTNNA1, BRCA2, STK11, SDHB, PRSS1, ATM, MSR1, PALB2 (PMID: 26182300).
The section about HDGC is now expanded, considering also cases with lack of CDH1 mutations, displaying alterations among other related genes, like: CTNNA1, BRCA2, STK11, SDHB, PRSS1, ATM, MSR1, PALB2.
2.) ALDH2 polymorphisms were reported to affect the development of gastric cancer associated with alcohol intake (e.g. PMID: 21507992), this could be discussed in the appropriate section.
The dependence between: “ALDH2 polymorphisms and alcohol intake affecting GC development” is discussed and added to the section “2.3. Risk Factors”.
3.) In Table 1. authors claim that "cyclin D1 and retinoblastoma protein (pRb) are positive regulators of the cell cycle process". This is not true for pRb. pRb acts as cell cycle repressor, it promotes G1/S arrest and growth restriction through inhibition of the E2F transcription factors. Please, correct this statement.
This mistake is corrected, the statement for pRb role is added.
4.) The choice of graphical representation of diagram 2. is a bit unclear to me. What should the arrow represent? The age of onset? But then it would make more sense to have the order in the other direction (youngest patients on the left, oldest on the right - and then please label somewhere that the arrow represents the age dimension). From the first glance on the arrow and without prior knowledge, the graphics more intuitively, albeit wrongly, suggest that conventional GC can somehow "progress" into early-onset GC. etc. Possibly authors could consider including a more intuitive diagram.
The diagram 2 is now changed for more intuitive; better understanding the importance of the percentage of the chosen GC subtype should be clearly visible in the current format.
Reviewer 2 Report
It was well written and educative. I enjoyed reading.
However, before publication, please add the role of imaging (ex, CT, PET or MR) and current status of targeted therapy for GC.
Author Response
Dear reviewer,
Thank you for a detailed review of our manuscript , " Gastric Cancer: Epidemiology, Risk Factors, Classification, Genomic Characteristics and Treatment Strategies". The manuscript has been improved taking into account the comments in your review. Our response point by point is given below.
It was well written and educative. I enjoyed reading
1.) However, before publication, please add the role of imaging (ex, CT, PET or MR) and current status of targeted therapy for GC
The current status on targeted therapies among GC patients is described in section 5.8.
The chapter 5.9. is added about imaging strategies in gastric cancer.